# Dectin-1 as a Potential Inflammatory Biomarker for Metabolic Inflammation in Adipose Tissue of Individuals with Obesity

**DOI:** 10.3390/cells11182879

**Published:** 2022-09-15

**Authors:** Ashraf Al Madhoun, Shihab Kochumon, Fatema Al-Rashed, Sardar Sindhu, Reeby Thomas, Lavina Miranda, Fahd Al-Mulla, Rasheed Ahmad

**Affiliations:** 1Animal and Imaging Core Facilities, Dasman Diabetes Institute, Dasman 15462, Kuwait; 2Genetics and Bioinformatics, Dasman Diabetes Institute, Dasman 15462, Kuwait; 3Immunology and Microbiology Department, Dasman Diabetes Institute, Dasman 15462, Kuwait

**Keywords:** dectin-1, obesity, metabolic inflammation, proinflammatory markers, adipose tissue

## Abstract

In obesity, macrophage activation and infiltration in adipose tissue (AT) underlie chronic low-grade inflammation-induced insulin resistance. Although dectin-1 is primarily a pathogen recognition receptor and innate immune response modulator, its role in metabolic syndromes remains to be clarified. This study aimed to investigate the *dectin-1* gene expression in subcutaneous AT in the context of obesity and associated inflammatory markers. Subcutaneous AT biopsies were collected from 59 nondiabetic (lean/overweight/obese) individuals. AT gene expression levels of *dectin-1* and inflammatory markers were determined via real-time reverse transcriptase-quantitative polymerase chain reaction. Dectin-1 protein expression was assessed using immunohistochemistry. Plasma lipid profiles were measured by ELISA. AT *dectin-1* transcripts and proteins were significantly elevated in obese as compared to lean individuals. AT *dectin-1* transcripts correlated positively with body mass index and fat percentage (r ≥ 0.340, *p* ≤ 0.017). AT *dectin-1* RNA levels correlated positively with clinical parameters, including plasma C-reactive protein and CCL5/RANTES, but negatively with that of adiponectin. The expression of *dectin-1* transcripts was associated with that of various proinflammatory cytokines, chemokines, and their cognate receptors (r ≥ 0.300, *p* ≤ 0.05), but not with anti-inflammatory markers. *Dectin-1* and members of the TLR signaling cascade were found to be significantly associated, suggesting an interplay between the two pathways. *Dectin-1* expression was correlated with monocyte/macrophage markers, including *CD16*, *CD68*, *CD86*, and *CD163*, suggesting its monocytes/macrophage association in an adipose inflammatory microenvironment. *Dectin-1* expression was independently predicted by *CCR5*, *CCL20*, *TLR2*, and *MyD88*. In conclusion, dectin-1 may be regarded as an AT biomarker of metabolic inflammation in obesity.

## 1. Introduction

The global prevalence of obesity has increased manifold over the last four decades [1,2]. Worldwide, the health community has been exploring preventive strategies for obesity, including nutritional and surgical interventions. Miscellaneous factors cause obesity [3]; thus, personalized medical treatments and healthcare policy changes addressing emerging needs could be more effective [4,5,6]. Obesity is a potential risk factor for other metabolic disorders [7,8]. Nonetheless, the intertwined roles of genetic and environmental cofounders remain to be further elucidated [9,10]. The complex pathophysiology of obesity implicates the alteration of critical factors, including glucose homeostasis, dyslipidemia, and blood pressure. However, chronic systemic low-grade inflammation is reportedly the fundamental and main cause [11,12]. Remarkably, the precise mechanisms of the association of inflammation with metabolic syndrome are yet to be identified [13,14].

Adipose tissue (AT) is composed of multiple cells of different origins. Besides adipocytes, AT contains progenitor cells, residential leukocytes, and neuronal and vasculature cells [15,16]. Communication between these cell types has been found to be crucial for maintaining the tissue homeostasis and an active response to changes in physiological and environmental conditions [17]. AT is also a dynamic endocrine organ that produces and secretes hormones, adipokines, and chemokines [18]. Aberrant changes in the AT are associated with inflammatory responses and metabolic dysfunctions observed in obese patients [19,20]. The onset of obesity is often associated with rapid AT expansion, causing inadequate vascularization, hypoxia, and inflammation. Dysfunctional AT releases high levels of free fatty acids, adipokines, and proinflammatory cytokines/chemokines in the bloodstream which could reach other organs and ultimately affect their harmonized functions [21]. Notably, the secretion of inflammatory markers is chronologically correlated with the stage of obesity and its associated complications including insulin resistance and cardiovascular diseases [22,23].

Dectin-1 belongs to the type II transmembrane family C-type lectin receptors (CLRs) expressed in myeloid cells (monocytes, macrophages, and neutrophils) and antigen-presenting dendritic cells [24,25] and is involved in fungal recognition and innate immune response modulation. Dectin-1 binds to β-(1,3)-glucans at the fungal cell wall and triggers proinflammatory cytokine stimulation through the activation of reactive oxygen species and NF-*κ*B signaling [26,27,28]. Dectin-1 deficiency in humans and rodent knockout models has been associated with a moderate, but noninvasive, fungal infection, suggesting the existence of compensatory *antifungal response* mechanisms [29,30,31]. As reported by Mata-Martinez et al. dectin-1 activation triggers numerous downstream cellular responses, including the production of proinflammatory mediators, induction of phagocytosis, and activation of cytotoxic T-cell responses through multifactorial mechanisms [32].

The emerging roles of dectin-1 as a positive regulator of AT inflammation in high-fat diet (HFD)-fed MyD88 KO mice and as a biomarker for metabolic dysregulation in humans have recently been reported [33]. Nonetheless, its multifunctional roles in the human subcutaneous AT remain elusive. In this study, we aimed to investigate the changes in gene expression of dectin-1 in relation to modulations in the inflammatory and insulin resistance markers in the human subcutaneous AT.

## 2. Material and Methods

### 2.1. Study Population and Anthropometric Measurements

This study cohort included 59 nondiabetic individuals from both sexes who were recruited in the study at the Dasman Diabetes Institute, Kuwait. The exclusion criteria included pregnant women and individuals with diseases, such as lung, heart, kidney, or liver complications, immune dysfunction, diabetes, cancers, or hematologic disorders as previously described [34,35,36]. Using the standard formula for the body mass index (BMI) (BMI = body weight (kg)/height^2^ (m^2^)), the cohort was divided into 10 lean (BMI < 25 kg/m^2^), 20 overweight (25 ≤ BMI < 30 kg/m^2^), and 29 obese (BMI ≥ 30 kg/m^2^) individuals. For each category, the sample size was dependent on the sample availability and each participant’s decision to be engaged in the research study. Written informed consents were obtained from all study participants in accordance with the ethical guidelines stipulated in the Declaration of Helsinki and approved by the ethics committee of Dasman Diabetes Institute, Kuwait (Grant numbers: RA 2010-003; June 2010). Weights and heights were measured using calibrated electronic weighing scales and height-measuring bars. Waist circumferences were measured using constant-tension tapes. The IOI 353 Body Composition Analyzer (Jawon Medical, Seoul, Korea) was used to determine the whole-body compositions (percent body fat, soft lean mass, and total body water).

### 2.2. Collection of Subcutaneous AT

As previously described [35], human AT biopsy samples (approximately 500 mg) were collected from abdominal subcutaneous fat pads located adjacent to the umbilicus using standard sterile surgical procedures [37]. Briefly, local anesthesia, i.e., 2% lidocaine, was applied to the periumbilical area post alcohol decontamination. Tissue biopsy samples were collected through a small superficial skin incision (5 mm). After removal, the fat tissue was further dissected into smaller pieces of approximately 50–100 mg, rinsed with cold phosphate buffered saline (PBS), preserved in RNAlater, and stored at −80 °C until use [38]. For immunohistochemistry (IHC) studies, fresh AT fragments were formalin fixed, paraffin embedded, and stored at room temperature till use as described in [38].

### 2.3. Measurement of Metabolic Inflammatory Markers

Peripheral blood was collected from fasted individuals and evaluated for metabolic and biochemical markers, as has been described previously [39,40,41]. Fasting blood glucose (FBG) and lipid profiles, including plasma triglycerides (TGL), high-density lipoprotein (HDL), low-density lipoprotein (HDL), and total cholesterol (Chol), were measured using a Siemens Dimension RXL chemistry analyzer (Diamond Diagnostics Holliston, MA, USA). HbA1c was measured using VARIANT II (Bio-Rad, Hercules, CA, USA). Insulin resistance, HOMA-IR, was calculated from basal FBG and insulin concentrations using the following formula: HOMA-IR = fasting insulin (μU/L) × fasting glucose (nmol/L)/22.5. Plasma high-sensitivity CRP levels were measured by ELISA (BioVendor, Ashville, NC, USA). All assays were performed following the instructions of the manufacturers. White blood cell (WBC) count was measured using hematocytometry.

### 2.4. RNA Extraction, cDNA Synthesis, and RT-qPCR Reactions

Human AT tissue total RNA was extracted using the RNeasy kit (Qiagen, Valencia, CA, USA) as described in the manufacturer’s protocol. The first strand cDNA was synthesized from 500 ng total RNA using a High Capacity cDNA Reverse Transcription kit (Applied Biosystems, Foster City, CA, USA) as previously described [42]. Real-time reverse transcriptase-quantitative polymerase chain reaction (RT-qPCR) was performed as described elsewhere [42]. Briefly, cDNA samples (50 ng) were amplified using TaqMan Gene Expression Master Mix (Applied Biosystems) and gene-specific 20× TaqMan gene expression assays (Applied Biosystems) containing appreciated primers for target genes (listed in Appendix A) and target-specific TaqMan MGB probe labeled with FAM dye at the 5′-end and NFQ-MGB at the 3′-end of the probe using the 7900 Fast Real-Time PCR System (Applied Biosystems). The RT-qPCR reaction was initiated with uracil-DNA glycosylases (UDG, 120 s at 50 °C) and AmpliTaq gold enzyme (10 min at 95 °C) activation cycles, followed by 40 cycles where each cycle involved denaturation (15 s at 95 °C) and annealing/extension (60 s at 60 °C). Relative gene expression to the lean AT control was also calculated using the comparative cycles to threshold (C_T_) method, as has been described previously [43]. Results were normalized using glyceraldehyde 3-phosphate dehydrogenase (*GAPDH*), and means ± standard error of the mean (SEM) are expressed as fold changes in the expression relative to controls as indicated [44].

### 2.5. Immunohistochemistry (IHC)

Paraffin-embedded sections (4 μm thick) of subcutaneous AT were deparaffinized in xylene and rehydrated through descending grades of ethanol (100%, 95%, and 75%) to water. Antigen retrieval was then performed by placing slides in a target retrieval solution (pH 6.0; Dako, Glostrup, Denmark) in the pressure cooker boiling for 8 min and cooling for 15 min. After washing in PBS, endogenous peroxidase activity was blocked with 3% H_2_O_2_ for 30 min and non-specific antibody binding was blocked with 5% nonfat milk for 1 hr, followed by 1% bovine serum albumin solution for 1 h. The slides were incubated at room temperature overnight with primary antibody (1:100 dilution of rabbit polyclonal anti-dectin-1 antibody (Abcam, Waltham, MA, USA; #ab140039)). After washing with PBS (0.5% Tween), the slides were incubated for 1 h with secondary antibody, namely goat anti-rabbit conjugated with horseradish peroxidase polymer chain DAKO EnVision Kit (Dako, Denmark), and color was developed using a 3,3‘-diaminobenzidine (DAB) chromogen substrate. The specimens were washed, counterstained, dehydrated, cleared, and mounted as described elsewhere [45]. For analysis, digital photomicrographs of the entire AT sections (20×; Pannoramic Scan, 3DHistech, Budapest, Hungary) were used to quantify the immunohistochemical staining using ImageJ software (NIH, Bethesda, MD, USA). Dectin-1 antibody specificity was validated using spleen tissue, as shown in Appendix A.

### 2.6. Statistical Analysis

Statistical analysis was performed using GraphPad Prism software (GraphPad, La Jolla, CA, USA) and SPSS for Windows version 19.01 (IBM SPSS Inc., Chicago, IL, USA). Unless otherwise indicated, data were shown as mean ± SD values. Since the sample size is small and the data are not normally distributed, non-parametric Mann-Whitney U test was used to compare means between groups. Spearman correlation and multivariable regression analysis were performed to determine associations between different variables. As has been previously described [44,46], in all analyses, a *p* < 0.05 was considered significant. Standard multivariable linear regression by the Enter method was used; variables that significantly correlated with dectin-1 were selected as predictor variables and were entered simultaneously to generate the model. The F-test was used to assess whether the set of entered independent variables collectively predicted the dependent variable. R-squared was used to determine how much variance in the dependent variable could be accounted for by the set of independent variables. The *t* test, *p* value, and beta coefficients (*β*-value) were used to determine the significance and the magnitude of prediction for each independent variable, respectively.

## 3. Results

### 3.1. Demographic and Clinical Characteristics of the Study Population

The physiological and biochemical characteristics of the 59 individuals from both genders included in this study are summarized in Table 1. All participants were between 32 and 58 years old with comparable heights. However, the overweight and obese individuals were significantly heavier than the lean individuals. Furthermore, in comparison to the lean participants, the overweight and obese individuals were determined to have statistically significantly greater BMI, waist circumference, body fat, and plasma triglyceride levels (Table 1).

The three groups had comparable levels of plasma cholesterol and LDL, unlike HDL, which, as expected, was significantly lower in overweight and obese individuals than in lean individuals (Table 1). In this study, diabetic patients were excluded. However, for all participants, the FBG and HbA1c measurements ranged from 4.3 to 6.1 mmol/L and from 5.0% to 6.3%, respectively, indicating that the population is at normal and/or prediabetic status. This was further supported by a statistically significant increase in HOMA-IR in the obese individuals, as compared to that of the lean individuals (Table 1). Furthermore, the physical and biochemical parameters did not reflect differences related to gender within the studied cohort (Appendix A).

### 3.2. Dectin-1 Gene Expression Is Associated with Obesity

*Dectin-1* has been shown to be elevated in mesenteric AT isolated from a small group of obese individuals [33]. To elaborate the prospective role of *dectin-1* in other adipose tissues, we have examined the *dectin-1* gene and protein expression profiles in the subcutaneous AT biopsies from healthy individuals with different BMIs. Relative to lean individuals, AT *dectin-1* transcripts and protein expressions were significantly increased in obese individuals as assessed by RT-qPCR and immunohistochemistry analyses, respectively (*p* < 0.009; Figure 1). In overweight individuals, we observed a relative increase in *dectin-1* mRNA (Figure 1A), while a statistically significant increase of dectin-1 protein was observed (*p* = 0.0143, Figure 1B,C). The minor differences in expression significancy between these molecules could be attributed to the sample size used in studying each molecule, as well as the sensitivity of the analytical techniques. Overall, the data indicate that AT *dectin-1* RNA and proteins are elevated as a function of obesity.

### 3.3. Increased AT Dectin-1 Gene Expression in Obesity Correlates with the Metabolic and Immune Markers

AT gene expression of *dectin-1* was found to correlate positively with the clinical indicators of obesity including BMI (r = 0.3557, *p* = 0.007; Figure 2A) and body fat percentage (r = 0.340, *p* = 0.017; Figure 2B), indicating its strong association with obesity. In our cohort, AT *dectin-1* transcripts expression was also found to be positively associated with systemic inflammatory biomarkers, including serum CRP (r = 0.363, *p* = 0.023; Figure 2C) and CCL5/RANTES (r = 0.334, *p* = 0.053; Figure 2D) levels. On the other hand, as expected, a negative correlation was observed between AT *dectin-1* transcripts and serum adiponectin protein levels (r = -0.505, *p* = 0.0017; Figure 2E). Together, these data suggest that dectin-1 may represent a biomarker for obesity and associated inflammation.

### 3.4. Increased Adipose Dectin-1 Gene Expression Is Associated with Inflammatory Signatures

Next, we determined the correlation of the transcriptional expression levels of *dectin-1* to those of inflammatory markers in AT. As shown in Table 2 and Appendix A, our data indicate that, in obese individuals, *dectin-1* RNA expression correlated positively with that of the proinflammatory markers including *IL-8*, *IL-18*, and *IL-23A* (r ≥ 0.549; *p* < 0.0002) and slightly positively correlated with *IL-1β* (r = 0.323; *p =* 0.033). Notably, *dectin-1* transcripts were also correlated with that of *IL-10*, which is a regulatory cytokine that exerts both pro- and anti-inflammatory (pleiotropic) effects. No correlations between *dectin-1* and *IL-2*, *IL-6*, or *IL-33* transcripts were observed (Table 2, and Appendix A). Moreover, no statistically significant correlation was observed between the transcripts of *dectin-1* and that of the B-cell maturation and differentiation markers *IL-5* or *IL-13*, nor with that of the T*h*1 cell marker *IL-12A*, suggesting that *dectin-1* is not associated with anti-inflammatory, T*h*1, or pleiotropic biomarkers. Moreover, the proinflammatory *TNF-α* transcription levels were positively correlated with those of *dectin-1* (r = 0.491, *p* = 0.0002; Table 2, and Appendix A).

Interestingly, *dectin-1* gene expression levels were positively associated with that of some closely related CC motif chemokines and their receptors mediating obesity-induced chronic inflammation, such as *CCL3*, *CCL8*, and *CCL20*, markers for macrophages and lymphocytes, as well as *CCL2* and *CCL7*, which are well-defined monocyte chemoattractants. Furthermore, the gene expression of *dectin-1* and *CCL5/RANTES* was also found to be positively correlated in AT (r = 0.527; *p* < 0.0001; Table 2, and Appendix A). Further, *dectin-1* gene expression was also found to be associated with that of C-X-C motif chemokines, particularly *CXCL10* and *CXCL11* (r ≥ 0.438; *p* < 0.001) and slightly positively correlated with that of *CXCL9* (r = 0.260; *p* = 0.049; Table 2). To sum up, the gene expression association studies reveal that the transcripts of *dectin-1* are significantly correlated with those of a subset of interleukins and chemokines, suggesting that *dectin-1* crosstalk is limited to certain cell populations localized within the heterogeneous AT.

### 3.5. Increased AT Dectin-1 Gene Expression in Obesity Is Associated with Toll-Like Receptors (TLRs), Downstream Signaling Molecules, and Inflammatory Leukocyte Subpopulations

TLRs are important modulators of the innate immune system and are known to play a crucial role in AT inflammation by activating expression of inflammatory cytokines and chemokines. Since dectin-1 action is mediated by IRF5 [47], which is regulated by TLRs, we hypothesized a prospective correlation between AT *dectin-1* expression and TLRs at transcriptional levels. Indeed, a significant correlation was noted between *dectin-1* mRNA levels and that of *TLR2*, *TLR7*, *TLR 8*, and *TLR10* (r values ≥ 0.423; *p* ≤ 0.001; Table 3, and Appendix A). No association was found between the *dectin-1* gene expression and that of *TLR3*, *TLR4*, or *TLR9*.

Further associations were observed with downstream effectors of the TLR signaling pathways. AT expression levels of *dectin-1* RNA were positively corelated with that of the innate immune signal transduction adaptor *MyD88* (r = 0.408; *p* = 0.002) and its associated IL-1R-associated kinase 1 (*IRAK1*) (r = 0.360; *p* = 0.007) expression. Importantly and as anticipated, gene expression levels of *dectin-1* and *IRF5* were also found to be positively correlated (r = 0.492; *p* = 0.0002; Table 3, and Appendix A).

AT contains residential immune cells, including T-cells, B-cells, dendritic cells, NK cells, and monocytes/macrophages, which become functionally dysregulated over time in the context of obesity [48]. *Dectin-1* gene expression levels have been found to be associated with distinct proinflammatory cytokines and chemokines that are secreted by different leukocyte subpopulations in AT. Toward this end, our data showed a positive correlation between *dectin-1* transcript levels and that of the common monocyte and macrophage markers *CD11c*, *CD16*, and *CD68*; M1 macrophages co-stimulatory marker *CD86*; and M2 macrophage immune sensing marker *CD163* (Table 3, and Appendix A). In addition, dectin-1 proteins were found to be colocalized with that of the M1 macrophage marker CD64, as well as that of the M2 marker CD163 (Appendix A, respectively).

### 3.6. Dectin-1 as an Independent Predictor of AT Inflammatory Markers

To determine which inflammatory parameter was independently correlated with the elevated AT *dectin-1* transcript levels in overweight/obese individuals, the parameters showing a significant association were included for a further multiple stepwise regression analysis. The multiple regression analysis revealed that *CCL20* (β = 0.024, *p* = 0.0004), *CCR5* (β = 0.803, *p* = 0.0002), *TLR2* (β = 1.754, *p* < 0.0001), and *MyD88* (β = 2.036; *p* = 0.019) were independently associated with *dectin-1* (Table 4).

## 4. Discussion

Dectin-1 is recognized as a pattern recognition receptor (PRR) expressed on myeloid cells (monocytes, macrophages, and neutrophils) and antigen-presenting dendritic cells. Whereas the expression of PRR such as TLRs is well studied in obesity, changes in dectin-1 expression in the adipose tissue in obesity and its significance remain largely unclear. Herein, we show the elevated *dectin-1* gene/protein expression in the subcutaneous adipose tissue in humans with obesity to be associated positively with BMI, body fat percentage, and CRP levels in those individuals. Similar observations have also been reported in the mesenteric (visceral) AT of obese individuals [33]. Moreover, expression of dectin-1 was found to be negatively correlated with adiponectin, which is a key adipokine involved in energy metabolism, with antidiabetic and anti-inflammatory properties [49,50]. Together, these data suggest a potential role of dectin-1 in obesity and metabolic syndromes.

AT is an active endocrine organ that secretes several hormones, cytokines, and chemokines, collectively known as adipokines [16], which sustain body homeostasis through dynamic processes, including nutrient intake, insulin sensitivity, and immunomodulation [51]. Understanding the role of AT adipokines and metabolites is fundamental to prevent abnormalities or dysfunction and maintain the metabolic homeostasis.

Leukocyte activation and trafficking to inflammatory sites is known to be regulated by proinflammatory factors secreted within the AT. Among obese individuals that we studied, the upregulation of dectin-1 was found to correlate with expression of proinflammatory mediators, including several interleukins, CC chemokines, CXC chemokines, and TNF-α, but not with anti-inflammatory factors or pleiotropic cytokines.

The interplay between dectin-1 and inflammatory mediators has been reported. Elevated levels of *dectin-1* transcripts were detected in monocytes isolated from diabetic patients, which was also associated with defective production of *IL-10* [52], a potent regulatory cytokine with pleiotropic functions and recently reported to have a proinflammatory role depending on the ambient conditions [53,54,55].

The examined proinflammatory mediators have been associated with the pathogenesis of various metabolic disorders, such as diabetes, heart diseases, and atherosclerosis. Spiegelman et al. were the first to demonstrate the association of TNF-α with AT insulin resistance in obesity using animal models [56]. It has been reported that the levels of IL-1β, IL-2, and IL-6 are augmented in AT samples from obese individuals [57,58]. Taken together, positive associations at the transcriptional level between dectin-1 and multiple inflammatory markers imply that in obesity, dectin-1 expression is induced in the inflamed AT. Moreover, significant associations between AT *dectin-1* expression and several proinflammatory mediators, especially the chemokines, indicate that these changes parallel the inflammatory cell infiltration into the AT, which might contribute to the pathogenesis of obesity, including the development of insulin resistance.

In obesity, white adipose tissue is expanded, including both hyperplasia and hypertrophy. The adipose tissue is both storage and an active metabolic organ. The increased metabolic activity within the expanding adipose tissue requires a consistent supply of oxygen as well as nutrients to suffice physiological needs and provide energy fuel. Therefore, adipose expression of the vascular endothelial growth factor (VEGF) triggers the process of angiogenesis to support neovascularization in the adipose tissue in obesity. These changes ensure a suitable microenvironment which is supportive of normal cellular metabolic activity and function. In our study, the angiostatic chemokines *CXCL9*, *CXCL10*, and *CXCL11* were found to be correlated with *dectin-1*. Whereas on one hand, these factors are sufficient to recruit and activate immune cells in the AT in obese individuals, on the other hand, their angiostatic effect might create a hypoxic tissue microenvironment, causing or exacerbating inflammation and metabolic dysfunction. Consistent with our study, at least in part, Hueso et al. observed an elevated gene expression of *CXCL10* and *CXCL11* in obese visceral AT, associated with the reduced neovascularization, suggesting that the impaired angiogenesis was most likely associated with increased expression of angiostatic chemokines [59].

TLRs are well-characterized as the innate immune receptors and lately as the nutrient sensors that mediate metabolic inflammation through chronic expression of proinflammatory mediators, particularly in obese AT. TLR activation triggers a dynamic transduction cascade mediated by MyD88 and IRAK proteins, secreting proinflammatory cytokines [60]. Herein, we have delineated the relationship between AT expression of *dectin-1*, TLRs, and their related downstream signaling molecules. To this end, we found that elevated AT *dectin-1* transcripts expression was positively associated with *TLR2*, *TLR7*, *TLR8*, and *TLR10* gene expression in obese patients. In line with this finding, several studies show that TLRs expression in the AT is elevated in obese individuals, with/without type 2 diabetes, and these alterations can be directly associated with BMI, cytokines/chemokines expression, and insulin resistance [38,40,45,61]. Being a PRR, *dectin-1* synergizes with the TLR pathway to induce the expression of inflammatory cytokines, including *TNF-α*, *IL-1β*, and *IL-12* [62,63,64]. In our study, an increase in the AT *dectin-1* transcripts levels was associated with the TLR-downstream signaling molecules, including *MyD88*, *IRAK1*, and *IRF5*. MyD88 is a key adaptor protein involved in the signaling of all TLRs (except TLR3) and IL1R, linking them to the IRAK family kinases and downstream IRF factors [65]. Recently, Castoldi et al. suggested a new role for MyD88 in metabolic syndromes and observed a notable upregulation of *dectin-1* expression in AT residential macrophages isolated from MyD88 knockout mice, which correlated with the exacerbation of obesity and increased insulin resistance. Furthermore, they showed that treating mice with a dectin-1 inhibitor decreased the number of M1 macrophages in the AT and ameliorated insulin resistance [33].

We have previously observed a positive correlation between the expression levels of *IRF5* and *dectin-1* in diabetic obese patients, but not in diabetic lean or overweight individuals [66], which is in alignment with our current observations and suggests that obesity, but not diabetes, might be implicated in increased expression of both *dectin-1* and *IRF5*. Consistent with our finding of an increased expression of the mentioned markers, similar changes in the AT expression of *MyD88*, *IRAK1*, and *IRF5* have been reported in individuals with obesity and/or type-2 diabetes [33,45,66]. Together, our data point to a plausible link between dectin-1 and TLR signaling pathways, leading to metabolic inflammation in the AT.

Our data support a positive association between *dectin-1* transcripts expression and that of monocytes/macrophage markers, including *CD16*, *CD68*, *CD86*, and *CD163*. In obesity, monocytic extravasation into the AT occurs and fat cell necrosis initiates an M2 to M1 macrophage phenotypic shift in the adipose tissue, which is regarded as one of the causes of metabolic inflammation associated with the disease [67]. M2 macrophages are characterized by the increased expression of dectin-1, CD206, CD163, CCR2, CXCR1, CXCR2, and MgL-1/2 [68]. M2 macrophages are considered anti-inflammatory, secrete high levels of IL-10 and TGF-β but low levels of IL-12, and are abundantly found in lean adipose tissue [69]. M1 macrophages are characterized by the elevated expression of major histocompatibility complex class II (MHC-II), CD80/CD86 costimulatory molecules, and CD68 [70]. M1 macrophages are pro-inflammatory, secrete high levels of IL-6, TNF-α, IL-1β, CXCL-9/10, IL-12, and IL-23 but low levels of IL-10, and are predominant in obese adipose tissue [71,72]. A strong correlation of the dectin-1 expression with pan monocyte/macrophage markers, including those used for M1 (CD68) and M2 characterization (CD163), suggests that dectin-1 might be constitutively co-expressed on cells of myeloid lineage, regardless of their M1/M2 polarization status. Besides, associating CD68 and CD163 to represent M1 and M2 macrophage subtypes, respectively, is also controversial since the CD68 has also been detected on cells other than macrophages and dendritic cells, such as lymphocytes, fibroblasts, stromal cells, and endothelial cells [73], all of which are increased in the inflamed adipose tissue. It is also suggested that macrophages may skew towards M1 profile via a relatively reduced CD163 expression [74].

This study remains limited by certain caveats and concerns. First, the complexity of the intercellular crosstalk between adipocytes and resident immune cells may affect the correlative dynamics between dectin-1 and proinflammatory markers in the AT microenvironment. Second, we cannot exclude the confounding effects of the presence of other immunomodulatory factors/agents present in the adipose tissue that we could not herein study. Third, the existence of genetic polymorphisms within the human cohorts may also impact the outcome of such correlative investigations. In any case, further studies involving larger and more diverse cohorts will be required to investigate how the *dectin-1* expression might be induced or modulated by *TLR2*, *MyD88*, *CCL20*, and/or *CCR5*, all of which have been identified as the independent predictors of dectin-1 adipose expression in obesity, as per our preliminary findings from this study.

## 5. Conclusions

AT *dectin-1* transcripts expression is positively associated with the gene expression of proinflammatory cytokines/chemokines, their cognate receptors, TLRs, and downstream signaling partners, as well as monocyte/macrophage markers (as described in the schematic Figure 3). Taken together, our results suggest that adipose dectin-1 upregulation may be regarded as a potential predictor of metabolic inflammation in an obesity setting.

## Figures and Tables

**Figure 1 cells-11-02879-f001:**
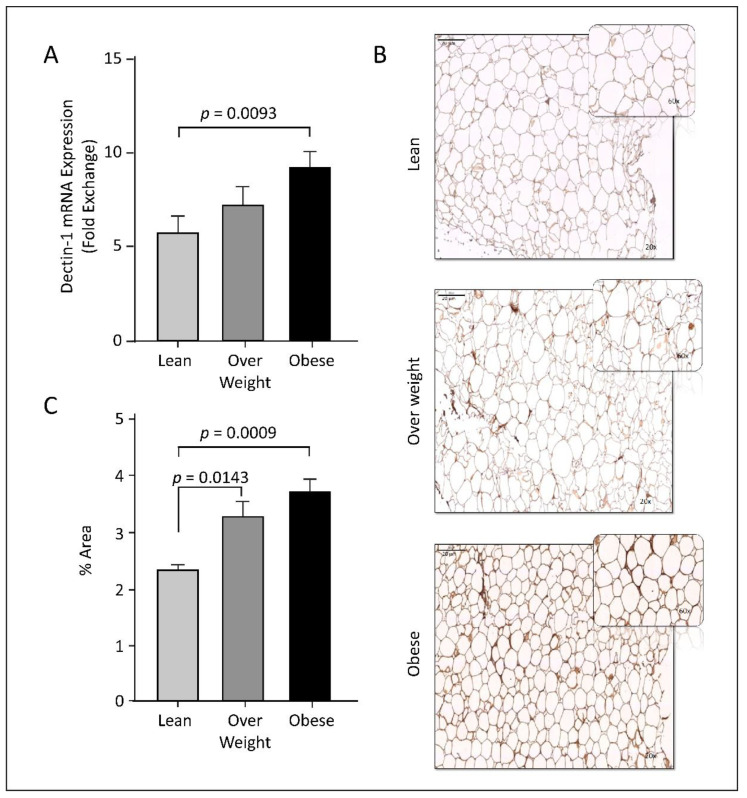
Upregulation of adipose tissue (AT) dectin-1 gene and protein expression in obesity. (**A**) *Dectin-1* gene expression was assessed in AT using qRT-PCR in 59 individuals grouped into lean, overweigh and obese based on their BMI, as described in Materials and Methods. *Dectin-1* transcripts expression was significantly increased in obese compared with lean adipose tissue (*p* = 0.0093). (**B**) Immunohistochemistry (IHC) analysis was performed to determine dectin-1 protein expression using five AT sections isolated from 5 participants per-group (Leans: 3 females + 2 males; Overweight: 1 females + 4 males; Obese: 4 males + 1 female). Screening of all IHC sections revealed no gender differences. Representative images for AT dectin-1 protein expression (magnification, 20×; enlarged area of image in top right corner, magnification, 60×) in obese, overweight, lean individuals. (**C**) Integrated optical density (IOD) divided by the adipocytes area was used to quantify dectin-1 protein expression in IHC sections of the adipose tissue samples from lean, overweight, and obese individuals. The data (mean ± SEM) show elevated dectin-1 protein expression in overweight (*p* = 0.0143) and obese (*p* = 0.0009) individuals compared to lean. Regarding IHC.

**Figure 2 cells-11-02879-f002:**
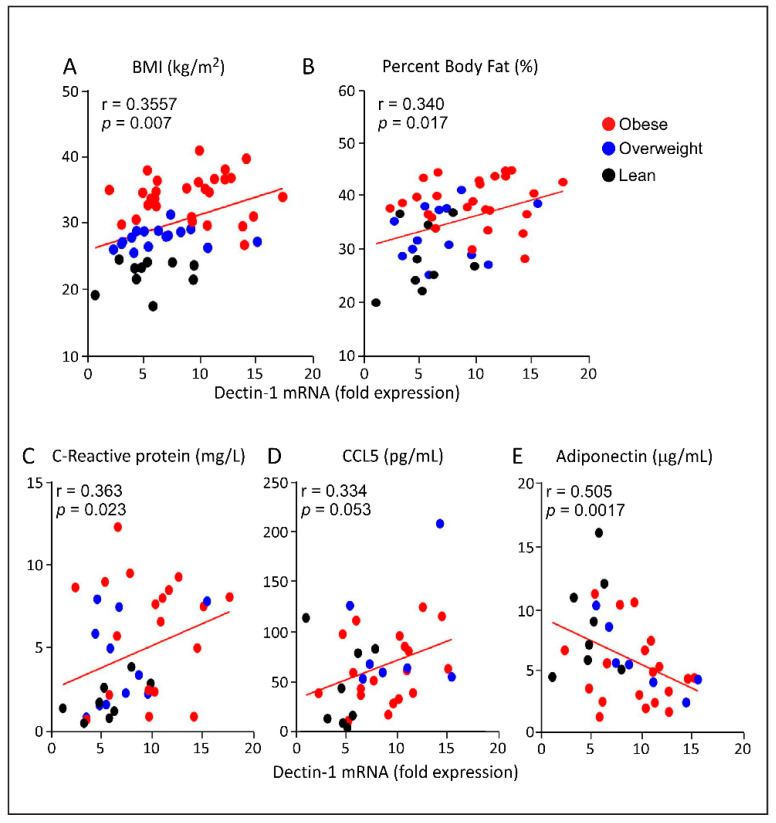
Correlation of the adipose tissue (AT) dectin-1 gene expression with metabolic and immune biomarkers. In our study cohort, we performed association studies between the levels of *dectin-1* transcripts in AT isolated from individual with different BMI and their clinical parameters as well as the inflammatory biomarkers related to obesity. *Dectin-1* gene expression was found to associate positively with (**A**) Body mass index (*p* = 0.007), (**B**) Percentage of body fat (*p* = 0.017), (**C**) Plasma C-reactive protein (*p* = 0.023), and (**D**) CCL5/RANTES (*p* = 0.053)). (**E**) On the other hand, AT *dectin-1* transcripts and plasma adiponectin were negatively correlated (*p* = 0.0017).

**Figure 3 cells-11-02879-f003:**
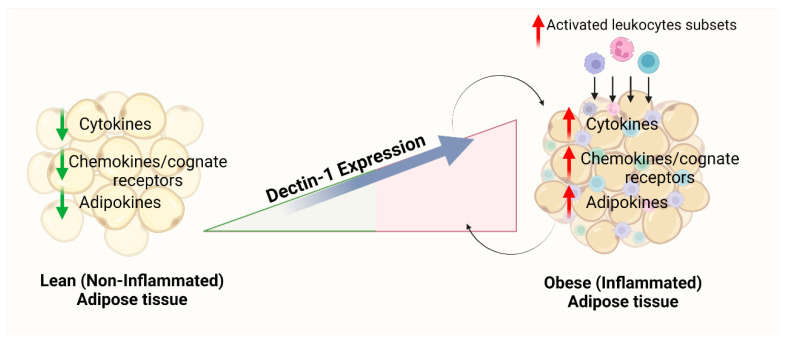
Schematic representation of dectin-1 and their association with metabolic inflammation in the context of obesity. Generated by BioRender software license number *PM247G7WOB*.

**Table 1 cells-11-02879-t001:** Demographic and clinical characteristics of the study population.

Metabolic Markers	Lean	Overweight	Obese	Lean vs. Overweight	Lean vs. Obese
	n = 10 (3M/7F) (Mean ± SD)	n = 19 (12M/7F) (Mean ± SD)	*n* = 30 (15M/15F) (Mean ± SD)	(*p*-Value)	(*p*-Value)
Age (years)	42.70 ± 8.17	43.68 ± 11.12	45.20 ± 13.12	0.807	0.572
Weight (kg)	62.93 ± 11.90	79.56 ± 9.92	94.48 ± 14.06	0.0004	<0.0001
Height (cm)	1.66 ± 0.12	1.68 ± 0.11	1.64 ± 0.11	0.68	0.754
BMI (kg/m^2^)	22.82 ± 2.35	28.27 ± 1.19	34.88 ± 3.22	<0.0001	<0.0001
Waist (cm)	81.33 ± 12.44	95.19 ± 8.81	107.15 ± 12.83	0.003	<0.0001
Body fat (%)	28.37 ± 6.27	32.52 ± 4.87	39.47 ± 4.28	0.073	<0.0001
FBG (mmol/L)	4.97 ± 0.64	5.26 ± 0.68	5.37 ± 0.76	0.282	0.14
TGL (mmol/L)	0.63 ± 0.24	1.19 ± 0.64	1.34 ± 0.85	0.002	<0.0001
Chol (mmol/L)	5.30 ± 1.11	4.98 ± 0.73	5.05 ± 1.14	0.348	0.544
HDL (mmol/L)	1.69 ± 0.51	1.27 ± 0.30	1.17 ± 0.24	0.009	0.01
LDL (mmol/L)	3.31 ± 0.93	3.18 ± 0.67	3.29 ± 1.01	0.677	0.963
HbA1c (%)	5.66 ± 0.46	5.49 ± 0.45	5.70 ± 0.65	0.35	0.857
HOMA-IR	1.40 ± 0.64	1.71 ± 0.98	4.40 ± 3.80	0.413	0.009
WBC	5.57 ± 1.60	6.11 ± 1.43	6.49 ± 1.97	0.379	0.206

BMI, body mass index; FBG, fasting blood glucose; TGL, triglyceride; Chol, cholesterol; HDL, high-density lipoprotein; LDL, low-density lipoprotein; HbA1c, glycated hemoglobin A1c; HOMA-IR, homeostatic model assessment for insulin resistance; WBC, white blood cells.

**Table 2 cells-11-02879-t002:** Correlation of AT *dectin-1* gene expression with that of various cytokines/chemokines and their cognate receptors.

Inflammatory Markers	Spearman Correlation		
	*r*-Value	*p*-Value	n
Interleukins			
IL-1β	0.323 *	**0.033**	44
IL-2	0.247	0.067	56
IL-5	−0.048	0.733	52
IL-6	0.183	0.195	52
IL-8	0.569 **	**<0.0001**	49
IL-10	0.564 **	**<0.0001**	55
IL-12A	0.188	0.227	43
IL13	−0.082	0.562	52
IL-18	0.496 **	**0.0002**	53
IL-23A	0.549 **	**<0.0001**	57
IL-33	−0.03	0.83	55
TNF-α	0.491 **	**0.0002**	53
Cytokine/chemokines receptors			
IL-2RA	0.100	0.454	58
CCR1	0.446 **	**0.001**	55
CCR2	0.470 **	**0.001**	51
CCR5	0.681 **	**<0.0001**	57
CC chemokine ligands			
CCL2	0.266 *	**0.05**	55
CCL3	0.541 **	**<0.0001**	54
CCL5	0.527 **	**0.0001**	47
CCL7	0.515 **	**<0.0001**	54
CCL8	0.215	0.138	49
CCL11	0.194	0.159	54
CCL15	0.029	0.831	57
CCL18	0.569 **	**<0.0001**	56
CCL19	0.24	0.072	57
CCL20	0.624 **	**<0.0001**	56
CXC chemokine ligands			
CXCL9	0.260 *	**0.049**	58
CXCL10	0.438 **	**0.001**	56
CXCL11	0.446 **	**0.001**	57

(* *p* ≤ 0.05, ** *p* ≤ 0.001 statistically significant).

**Table 3 cells-11-02879-t003:** Correlation of AT *dectin-1* transcripts with that of the Toll-like Receptors (TLRs), downstream singling molecules and AT resident monocyte/macrophage markers.

Inflammatory Markers	Spearman Correlation		
	*r*-Value	*p*-Value	n
TLRs and downstream signaling markers			
TLR2	0.663 **	**0.0001**	50
TLR3	0.229	0.106	51
TLR4	0.054	0.718	48
TLR7	0.485 **	**0.0001**	58
TLR8	0.652 **	**0.0001**	55
TLR9	−0.073	0.59	57
TLR10	0.423 **	**0.001**	54
MyD88	0.408 **	**0.002**	57
IRAK1	0.360 **	**0.007**	55
IRF3	0.157	0.286	48
IRF5	0.492 **	**0.0002**	54
Monocyte/macrophage surface markers			
CD16	0.719 **	**<0.0001**	56
CD68	0.502 **	**<0.0001**	57
CD86	0.679 **	**<0.0001**	55
CD163	0.603 **	**<0.0001**	57

(** *p* ≤ 0.001 statistically significant).

**Table 4 cells-11-02879-t004:** Multi Linear Regression analysis, with Dectin-1 as a dependent variable.

ANOVA (Sig) R2 = 0.55; *p* < 0.0001		
Predictor Variable	Scandalized Confinement (β)	*p*-Value
CCR5	0.803	**0.0002**
CCL20	0.024	**0.0004**
TLR2	1.754	**<0.0001**
MyD88	2.036	**0.019**

## Data Availability

The data are available upon request.

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
