# Peer review of "Dectin-1 as a Potential Inflammatory Biomarker for Metabolic Inflammation in Adipose Tissue of Individuals with Obesity"

_cells, 2022, doi:10.3390/cells11182879_

Round 1
Reviewer 1 Report
In this interesting study, Madhoun et al. described the correlation between dectin-1 levels within adipose tissue with metabolic and proinflammatory parameters. My primary concerns are the novelty of the findings reported herein, once Castoldi et al. have reported similar results in adipose tissue visceral depots. Furthermore, this is a very descriptive study, and some important questions remain open. For example, is dectin-1 levels increased in macrophages or other immune cells resident in adipose tissue? The authors should widen this study and look at macrophages within AT subcutaneous depots.
Minor points:
- Overall the presentation of the results in the abstract section should be rewritten;
- Authors are highly encouraged to discuss further why mRNA dectin-1 levels were similar between lean and overweight individuals (Fig. 1A); meanwhile, immunohistochemistry showed an increased % area in overweight individuals (Fig. 1B; p =0.0143).
- Once authors performed both mRNA expression and protein expression analysis, this information must be highlighted in the results description section. Such information is unclear throughout the manuscript (fig. 2; table 2).
- Are dectin-1 levels correlated with adipocytes area?
Major point:
- In figure 3, the authors describe a correlation between dectin-1 levels and M1/M2 macrophage markers. This is a very controversial result. Are these macrophages presenting a hybrid phenotype? This information should be extensively discussed throughout the discussion section.
Author Response
Comments and Suggestions for Authors
Reviewer’s comment: In this interesting study, Madhoun et al. described the correlation between dectin-1 levels within adipose tissue with metabolic and proinflammatory parameters. My primary concerns are the novelty of the findings reported herein, once Castoldi et al. have reported similar results in adipose tissue visceral depots. Furthermore, this is a very descriptive study, and some important questions remain open. For example, is dectin-1 levels increased in macrophages or other immune cells resident in adipose tissue? The authors should widen this study and look at macrophages within AT subcutaneous depots.
Authors’ response: We thank the reviewer for realizing our study objectives and for positive comments. The pioneering study by Castoldi et al. (cited in our manuscript) establishes the link between dectin-1 and inflammation, insulin resistance and metabolic dysregulation. The authors reported that dectin-1 antagonism improved glucose homeostasis and reduced the CD11c+ adipose tissue macrophages in chow- and HFD-fed MyD88 KO mice. The authors also included data from visceral (mesenteric) adipose tissue samples obtained from 6 lean and 7 obese patients in their study; suggesting that dectin-1 might have therapeutic implications as a biomarker for metabolic inflammation/ dysregulation in humans. In our study, we measured dectin-1 expression in the subcutaneous adipose tissue samples obtained from a larger population comprising of 59 individuals, further classified based on BMI as lean, overweight, and obese. The increased dectin-1 expression in obese adipose tissue associated positively with several inflammatory markers, suggesting that dectin-1 could be a potential biomarker for metabolic inflammation in human obesity, and these conclusions are aligned with the findings of Castoldi et al.
As regards the comment that our study is descriptive and some important questions remain open, indeed, there is a paucity of data as we speak linking the altered adipose dectin-1 expression to metabolic consequences in setting of obesity. Whereas our data point to correlative evidence that the increased dectin-1 expression in obese adipose tissue concurs with typical inflammatory markers in this compartment, we agree that several questions still remain to be addressed. For instance, it remains unclear how is dectin-1 expression modulated in obesity progression with regard to major effector cell populations in the adipose tissue such as monocytes, macrophages, dendritic cells, T-cells and NK cells etc. It will be also interesting to investigate how will dectin-1 expression be modulated on adipocytes in obesity setting? Our unpublished data support that dectin-1 is also expressed by human adipocytes.
As kindly asked for, we have now included new data showing that Dectin-1 is expressed on the immune cell membrane. (Please see Supplementary Figure 3A-C)
Minor points
Reviewer’s comment 1: Overall the presentation of the results in the abstract section should be re-written
Author response 1: As kindly advised, we have re-written results in the abstract section, which has clearly improved the data comprehension and overall readability of the manuscript (Please see Abstract section, Page 1).
.
Reviewer’s comment 2: Authors are highly encouraged to discuss further why mRNA dectin-1 levels were similar between lean and overweight individuals (Fig. 1A); meanwhile, immunohistochemistry showed an increased % area in overweight individuals (Fig. 1B; p =0.0143).
Author response 2: We thank the reviewer for raising this up. We speculate that his difference between mRNA and protein analyses of dectin-1 might be due to the varying sample sizes regarding gene and protein expression studies. It may also be noted that the analytical methods of qRT-PCR and immunohistochemistry herein used for mRNA and protein detection of dectin-1, respectively, differ by assay sensitivity. (Please, see Page 6; line 219-222).
Reviewer’s comment 3: Once authors performed both mRNA expression and protein expression analysis, this information must be highlighted in the results description section. Such information is unclear throughout the manuscript (fig. 2; table 2).
Authors’ response 3: Thanks for bringing this up. In ‘Results’ section, we clarified that the correlation studies were based on dectin-1 expression at the transcriptional level, relative to those of other adipose tissue biomarkers or in relation to plasma markers (Please, see the highlighted correction in red color font).
Reviewer’s comment 4: Are Dectin-1 levels correlated with adipocytes area?
Author response 4: Thanks for the comment. Indeed, dectin-1 protein expression levels are correlated with the adipocytes area. These data are presented in Figure 1C (which was previously mislabeled at Y-axis). Now labeled correctly, the Y-axis represents the dectin-1 integrated optical density (IOD) over the adipocytes area.
Major point
Reviewer’s comment 5: In Table 3, the authors describe a correlation between dectin-1 levels and M1/M2 macrophage markers. This is a very controversial result. Are these macrophages presenting a hybrid phenotype? This information should be extensively discussed throughout the discussion section.
Author response 5: We found that dectin-1 transcripts expression correlated strongly with monocytes/macrophage markers such as CD16, CD68, CD86, and CD163. CD16 is a monocytic marker and CD86 is a costimulatory receptor. Generally, CD68 is recognized as an M1 macrophage (inflammatory) marker while CD163 is known as an M2 macrophage (anti-inflammatory) marker. In obesity, an M2 to M1 polarization shift occurs which is regarded as an underlying factor of obesity-associated metabolic inflammation. In our study, a strong correlation of the dectin-1 expression with pan monocyte/macrophage markers, including CD68 and CD163, suggests that dectin-1 might be co-expressed with myeloid lineage markers, regardless of M1/M2 profile. Notably, associating the CD68 and CD163 to M1 and M2 macrophage subtyping, respectively, remains controversial as the CD68 has also been detected on cells other cells of myeloid lineage, such as lymphocytes, fibroblasts, stromal cells, and endothelial cells, all of which are known to be elevated in the inflamed adipose tissue. It is also suggested that macrophages may skew towards the M1 profile via a relatively reduced CD163 expression. This information has been now discussed in the revised discussion section of the manuscript (Please see Page: 12-13; lines: 413-433).

Reviewer 2 Report
In the manuscript by Madhoun et al “Increased Adipose Tissue Expression of Dectin-1 in Obesity: Association with Metabolic Inflammation (cells-1860922) submitted to “Cells” the authors assessed the prospective role of determining dectin-1 expression in subcutaneous adipose tissues of healthy individuals with different BMIs as inflammatory biomarker to predict the level of obesity-induced dysregulation.
The topic is very interest to the scientific community. Unfortunately, the paper is hampered by very poor English which makes it for a laborious and confusing read.
Furthermore, the paper tends to gloss over relevant points, and shows very little critical acumen regarding the material introduce and fails to address relevant points. Authors should improve the integration of the measured parameter.
The study in based in associations. No causation or mechanistic links between development of obesity, dectin-1 and the other parameters was provided. The authors reported several moderate and some strong correlations but without any very strong correlation.
Methods: I have a concern about the statistics. The Unpaired two-tailed Student’s t-can be use when comparing two independent groups with equal variance. This study includes three groups. One of the more common statistical tests for three or more data sets is the Analysis of Variance, or ANOVA. However, to use this test, the data must meet certain criteria, such as the data should be normally distributed, also known as the Gaussian distribution. Nevertheless, no information was provided on the distribution of the data. If these assumptions are met, the ANOVA test would be an appropriate test to analyze the variance of a single dependent variable across the three data sets. With the correct analysis there may be a slight change in the results…
Methods: “This study cohort included 59 nondiabetic individuals from both sexes who were recruited”. No forward information about sex distribution is provide. It should at least mention the sample size by sex in each group. Ideally, sex would also be considered a variable in the statistical analysis.
Please mention how was collected the subcutaneous AT for the Immunohistochemistry in the “2.2 Collection of subcutaneous AT”
Table 1: It would be interest to have that information broken down by sex within each group.
Figure 1: No rational was provided for the use of 15 participants in immunohistochemistry (IHC). All data should be expressed to clearly show the sample size used, please provide that information in the figure. No information about the sex on the 5 individuals is provide, that should be added to the legend.
The word “participates” should be replace by participants.
Figure 2: it would be pertinent to be able to distinguish the 3 groups in it, either by the use of different symbols or colors.
Please provide the measured levels of the metabolic parameters used in Figure 2, Table 2, Table 3 and Table 4 for the different groups.
Discussion: please clarify the sentence “Angiogenesis is crucial for maintaining normal AT metabolic function and for effectively providing expanding AT with increased supply of oxygen and nutrients.”
Conclusion: please close the parentheses in the sentence “(as described in the schematic Figure 3 “
Author Response
In the manuscript by Madhoun et al “Increased Adipose Tissue Expression of Dectin-1 in Obesity: Association with Metabolic Inflammation (cells-1860922) submitted to “Cells” the authors assessed the prospective role of determining dectin-1 expression in subcutaneous adipose tissues of healthy individuals with different BMIs as inflammatory biomarker to predict the level of obesity-induced dysregulation.
Reviewer’s comment 1: The topic is very interest to the scientific community. Unfortunately, the paper is hampered by very poor English which makes it for a laborious and confusing read.
Author response 1: Thanks for approving of the work done. As regard linguistic concerns, the revised manuscript has been extensively edited for the English language errors and omissions and we now hope that the revised version will meet with the reviewer’s kind approval.
Reviewer’s comment 2: Furthermore, the paper tends to gloss over relevant points, and shows very little critical acumen regarding the material introduce and fails to address relevant points. Authors should improve the integration of the measured parameter.
Author response 2: We agree with the critical criticism of the reviewer. In addressing these concerns, extensive modifications have been done throughout the manuscript including English language corrections, sentence re-phrasing for effective communication, synthesis of information based on the data presented as well as referencing to the available literature for review and assessment. We hope the revised version reads much better.
Reviewer’s comment 3: The study in based in associations. No causation or mechanistic links between development of obesity, dectin-1 and the other parameters was provided. The authors reported several moderate and some strong correlations but without any very strong correlation.
Author response 3: We agree with the reviewer that our study is correlative and descriptive since little is known about dectin-1 changes in the subcutaneous fat and its implications in the human obesity. In the current study, we collected subcutaneous adipose tissue biopsies from a total of 59 individuals classified as lean, overweight, and obese. A detailed biochemical and clinical data have been presented from this cohort. Adipose tissue samples were analyzed for gene/protein expression of dectin-1 as well as gene expression of a wide array of inflammatory markers was assessed and compared with dectin-1 transcripts expression. We also identified that TLR2, MyD88, CCL20, and CCR5 independently predicted the dectin-1 expression in the adipose tissue, which paves way for further in-depth studies to investigate their causal relationship with altered adipose tissue expression of dectin-1 in obesity. Nonetheless, our preliminary studies lead to some interesting questions and thus provide leads for further studies to be carried out for knowledge advancement in the field. The study limitations have been enumerated in the revised discussion section.
Reviewer’s comment 4: Methods: I have a concern about the statistics. The Unpaired two-tailed Student’s t-can be use when comparing two independent groups with equal variance. This study includes three groups. One of the more common statistical tests for three or more data sets is the Analysis of Variance, or ANOVA. However, to use this test, the data must meet certain criteria, such as the data should be normally distributed, also known as the Gaussian distribution. Nevertheless, no information was provided on the distribution of the data. If these assumptions are met, the ANOVA test would be an appropriate test to analyze the variance of a single dependent variable across the three data sets. With the correct analysis there may be a slight change in the results.
Author response 4: The statistical analyses have been re-done for necessary corrections as kindly advised (Please see Page: 5, Line 179-193).
Reviewer’s comment 5: Methods: “This study cohort included 59 nondiabetic individuals from both sexes who were recruited”. No forward information about sex distribution is provide. It should at least mention the sample size by sex in each group. Ideally, sex would also be considered a variable in the statistical analysis.
Author response 5: Thanks for the comments and concerns. Accordingly, sample size by gender has been shown as well as dectin-1 associations with the inflammatory markers have been re-assessed after gender stratification. Please, Page 5, lines 202-204; and the provided Supplementary Table 2, and Supplementary Table 3, after adjusting to gender.
Reviewer’s comment 6: Please mention how was collected the subcutaneous AT for the Immunohistochemistry in the “2.2 Collection of subcutaneous AT”
Author response 6: We thank the reviewer for the comment. In addressing this concern, tissue collection and preparation procedures for the IHC have been now included the methodology section 2.2 (Please see Page: 4; lines 122-124)
Reviewer’s comment 7: Table 1: It would be interest to have that information broken down by sex within each group.
Author response 7: Having the information broken down by gender within each group will limit the correlation analysis as the number of data-sets will be too small to get conclusive results. We included these data at Supplementary Table 2, and Supplementary Table 3, after adjustment to gender.
Reviewer’s comment 8: Figure 1: No rational was provided for the use of 15 participants in immunohistochemistry (IHC). All data should be expressed to clearly show the sample size used, please provide that information in the figure. No information about the sex on the 5 individuals is provide, that should be added to the legend. The word “participates” should be replace by participants.
Author response 8: We thank the reviewer for the suggestions made. In compliance, we have now modified the legend to Fig 1 to include the requested information (Please see page 6-7; lines 228-237). As for IHC, randomly selected 5 sections per individual were examined, and 5 individuals comprised each group and the images from 10 different fields per sample were analyzed. All in all, this approach provided the required data strength for statistical analysis.
Reviewer’s comment 9: Figure 2: it would be pertinent to be able to distinguish the 3 groups in it, either by the use of different symbols or colors.
Author response 9: Done as requested (Please see Figure 1, Page 7).
Reviewer’s comment 10: Please provide the measured levels of the metabolic parameters used in Figure 2, Table 2, Table 3 and Table 4 for the different groups.
Author response 10: As per the institutional ethical regulations’ national guidelines, these data cannot be included.
Reviewer’s comment 11: Discussion: please clarify the sentence “Angiogenesis is crucial for maintaining normal AT metabolic function and for effectively providing expanding AT with increased supply of oxygen and nutrients.”
Author response 11: We clarified the sentence in more detailed discussion about angiogenesis. (Please see Page 14; Lines 438-445.
Reviewer’s comment 12: Conclusion: please close the parentheses in the sentence “(as described in the schematic Figure 3 “
Author response 12: Done.

Reviewer 3 Report
INCREASED ADIPOSE TISSUE EXPRESSION OF DECTIN-1 IN OBESITY: ASSOCIATION WITH METABOLIC INFLAMMATION
Madhoun A, Kochumon S at al. (Ahmad R Group)
The manuscript by Madhoum, Kochumon, and collaborators has presented data showing that the expression of dectin-1 is significantly higher in the adipose tissue of obese patients compared with lean ones. The increased mRNA and protein levels of dectin-1 in subcutaneous AT of obese subjects correlate with increased BMI fat accumulation and the higher gene expression of plasma and AT inflammatory markers (cytokine/chemokines/their receptors/ TLR2/7/8; IRAK MyD88, IRAK1, and IRF5). Dectin1 negatively correlates with adiponectin.
Besides its critical role against invading pathogens on mucosal surfaces, the C-type lectin receptor Dectin-1 ( a member of the PRR family of CLRs) was described to have increased expression in adipose tissue (AT) of obese individuals and induced AT macrophages polarization towards M1 profile, contributing to the inflammatory response during obesity and associated insulin resistance (Castoldi et al., 2017).
The research group headed by Dr R. Ahmad showed in a recent publication (Kochumon, Madhoun et al., 2020) that the subcutaneous AT of obese individuals exhibited high levels of CXCL chemokines ( particularly CXCL11), which was closely associated with BMI; fat mass; higher expression of inflammatory markers (cytokines, chemokines and their receptors, TLRs and associated proteins); and M1 markers. They suggested that this picture would be related to the development of inflammation and progression of insulin resistance in obese individuals.
In the present manuscript, the same authors analyzed Dectin-1 gene and protein expression in subcutaneous AT of individuals from the same cohort used in their last work. They correlated the results with the same inflammatory markers evaluated in their last work.
Interestingly, Dectin-1 gene expression was associated with CXCL10 and CXCL11.
The authors proposed dectin-1 as a “potential AT inflammatory biomarker and a prospective therapeutic target in obesity”. The data support those Castoldi and colleagues (2017) obtained, proposing that Dectin-1 “has therapeutic implications as a biomarker for metabolic dysregulation in humans”.
The data is complementary to others and may be used for other correlation points, especially with metabolism markers. Results should be carefully interpreted once there are only mRNA/ PCR assays for all mediators used for correlation. Data were poor explored in the discussion that was quite repetitive of the results, missing a more elaborate discussion on the main findings.
We are raising some issues that should be addressed:
1.The results on the AT samples from overweight (OVW)subjects were quite a few used throughout the work and poorly discussed. The weight and BMI of obese and OVW did not present significant statistical differences between them, being both different to the lean
In the Results section authors mentioned “a marginal surge of dectin-1 mRNA in overweight individuals”. The correlation between dectin-1 x BMI (Fig 2) seems to be similar (roughly) for both. Do other TLRs or inflammatory mediators behave similarly to dectin-1 in this?
2.Protein quantification (western blotting) can help observe more significant differences among lean-overweight and obese AT levels of dectin-1
3.Leptin expression does not vary? Correlates with dectin?
4.We suggest including the graphics (showing dots distribution- similar to Fig 2) for the main correlations between dectin-1 and inflammatory mediators as a supplementary figure. Readers better visualize it
5.The last work from the group point to the increased expression of CXCL angiostatic chemokines that were (now) found to be correlated with dectin-1.Vimentin, which has a role in angiogenesis and adipocytes homeostasis, is a ligand of dectin-1. Recent works showed that deletion of vimentin improves IR in obese. The higher number/ area of adipocytes in obese AT may justify the increased expression of dectin. Did the authors find any difference in vimentin or another angiogenic marker?
6.Discussion can be improved to avoid repeating the results
Author Response
INCREASED ADIPOSE TISSUE EXPRESSION OF DECTIN-1 IN OBESITY: ASSOCIATION WITH METABOLIC INFLAMMATION
Madhoun A, Kochumon S at al. (Ahmad R Group)
The manuscript by Madhoum, Kochumon, and collaborators has presented data showing that the expression of dectin-1 is significantly higher in the adipose tissue of obese patients compared with lean ones. The increased mRNA and protein levels of dectin-1 in subcutaneous AT of obese subjects correlate with increased BMI fat accumulation and the higher gene expression of plasma and AT inflammatory markers (cytokine/chemokines/their receptors/ TLR2/7/8; IRAK MyD88, IRAK1, and IRF5). Dectin1 negatively correlates with adiponectin.
Besides its critical role against invading pathogens on mucosal surfaces, the C-type lectin receptor Dectin-1 ( a member of the PRR family of CLRs) was described to have increased expression in adipose tissue (AT) of obese individuals and induced AT macrophages polarization towards M1 profile, contributing to the inflammatory response during obesity and associated insulin resistance (Castoldi et al., 2017).
The research group headed by Dr R. Ahmad showed in a recent publication (Kochumon, Madhoun et al., 2020) that the subcutaneous AT of obese individuals exhibited high levels of CXCL chemokines (particularly CXCL11), which was closely associated with BMI; fat mass; higher expression of inflammatory markers (cytokines, chemokines and their receptors, TLRs and associated proteins); and M1 markers. They suggested that this picture would be related to the development of inflammation and progression of insulin resistance in obese individuals.
In the present manuscript, the same authors analyzed Dectin-1 gene and protein expression in subcutaneous AT of individuals from the same cohort used in their last work. They correlated the results with the same inflammatory markers evaluated in their last work.
Interestingly, Dectin-1 gene expression was associated with CXCL10 and CXCL11.
The authors proposed dectin-1 as a “potential AT inflammatory biomarker and a prospective therapeutic target in obesity”. The data support those Castoldi and colleagues (2017) obtained, proposing that Dectin-1 “has therapeutic implications as a biomarker for metabolic dysregulation in humans”.
Reviewer’s comments: The data is complementary to others and may be used for other correlation points, especially with metabolism markers. Results should be carefully interpreted once there are only mRNA/ PCR assays for all mediators used for correlation. Data were poor explored in the discussion that was quite repetitive of the results, missing a more elaborate discussion on the main findings. Improve the discussion Section.
Author response: We thank the reviewer for kind interest in our work published over the past few years. The constructive comments and suggestions will definitely improve scientific merit of the manuscript. We agree with the reviewer that results based on correlative evidence when the expression analysis are made at the transcriptional level, warrant caution. In this regard, both Results and Discussion sections in the revised manuscript have been accordingly modified. Also, the major study limitations have been included at the end of the discussion section to warrant caution and suggest the lines of work for future studies.
Reviewer’s comment 1: We are raising some issues that should be addressed: The results on the AT samples from overweight (OVW) subjects were quite a few used throughout the work and poorly discussed. The weight and BMI of obese and OVW did not present significant statistical differences between them, being both different to the lean.
Author response 1: We did not do any comparison studies between OVW and obese individuals. In the data presented, we show comparisons between obese and lean as well as between OVW and lean. Thus, both groups were compared with a control (lean) group.
Reviewer’s comment 2: In the Results section authors mentioned “a marginal surge of dectin-1 mRNA in overweight individuals”.
Author response 2: Dectin-1 transcripts expression in AT samples from overweight individuals, relative to leans, was relatively higher which did not reach the statistical significance (Fig 1A). A relatively small sample size in overweight group could possibly be a reason. However, a significant difference of dectin-1 protein expression was observed between the two groups. Notably, the assays used for gene and protein expression differ by sensitivities of detection (Please see Page 6, Lines 222-225).
Reviewer’s comment 3: The correlation between dectin-1 x BMI (Fig 2) seems to be similar (roughly) for both.
Author response 3: The correlation analysis was done on data from all 59 individuals.
Reviewer’s comment 4: Do other TLRs or inflammatory mediators behave similarly to dectin-1 in this?
Author response 4: We did not study associations between the BMI and the TLRs or other inflammatory markers.
Reviewer’s comment 5: Protein quantification (western blotting) can help observe more significant differences among lean-overweight and obese AT levels of dectin-1
Author response 5: Western blot was not used for dectin-1 protein detection, due basically to the limited biopsy tissue availability. Therefore, IHC was used detectin-1 protein assessment in the adipose tissue while immunofluorescence was used to measure CD64 and CD163 expression (Supplementary Figure 3A-C).
Reviewer’s comment 6: Leptin expression does not vary? Correlates with dectin?
Author response 6: We found a positive correlation between dectin-1 gene expression and the plasma leptin levels; however, these data were obtained from 11 participants only and were, therefore, not included in the study.
Reviewer’s comment 7: We suggest including the graphics (showing dots distribution- similar to Fig 2) for the main correlations between dectin-1 and inflammatory mediators as a supplementary figure. Readers better visualize it.
Author response 7: Thanks for the suggestions. Accordingly, we have included the dot plots for all the factors that are correlated with dectin-1 expression (Please see the Supplementary Table 2, and the result section). Please note that we included all the datasets that we have, and we did not exclude any data point to avoid data selection bias.
Reviewer’s comment 8: The last work from the group point to the increased expression of CXCL angiostatic chemokines that were (now) found to be correlated with dectin-1. Vimentin, which has a role in angiogenesis and adipocytes homeostasis, is a ligand of dectin-1. Recent works showed that deletion of vimentin improves IR in obese. The higher number/ area of adipocytes in obese AT may justify the increased expression of dectin. Did the authors find any difference in vimentin or another angiogenic marker?
Author response 8: Thanks for the comments on angiostatic CXCLs that we earlier reported and the angiogenic vimentin reported by others as an endogenous ligand for dectin-1. We did not measure vimentin expression in the adipose tissue in obesity, but it will be worth investigating in future studies.
Reviewer’s comment 9: Discussion can be improved to avoid repeating the results.
Author response 9: Done as suggested.

Reviewer 4 Report
Ashraf Madhoun and colleagues submitted a manuscript entitled “Increased Adipose Tissue Expression of Dectin-1 in obesity: Association with Metabolic Inflammation” for publication as an original article in Cells, MDPI. Current study investigated in adipose tissue samples from 59-human subjects (non-diabetic) with obesity. The authors have investigated the expression of Dectin-1 in individuals with obesity and assessed that Dectin-1 transcript expression is positively correlated with higher transcript levels of pro-inflammatory markers related macrophage infiltration, cytokine and chemokines ligands and their receptors. The study design is appropriate and manuscript written very well, but several issues raise concerns.
Comments:
1. AS manuscript title appears like Adipose tissue by itself express the Dectin-1, as authors also mentioned in the introduction, Dectin-1 is transmembrane protein expressed in monocytes, Macrophages and other inflammatory cells. Therefore, I strongly suggest/recommend to authors to change the title of current manuscript and should be more realistic and meaningful. For Example: Dectin-1 acts as potential Inflammatory Biomarker for metabolic inflammation in Adipose tissue of individuals with Obesity.
2. As it known from previous publication (cell reports 2017), MyD88 is negatively corelated with Dectin-1 expression levels. In current study, do you see the regulation of MyD88 is increased or decreased? Authors need to show MyD88 transcript expression in bar graphs together with Figure1.
In Figure 1 Legend needs to reformat and rewrite including the Figure 1 title. The description in Figure 1 legend is very confusing Because authors start with describing Figure 1A, then suddenly immediately jumped to 1B before even finish describing 1A. Authors need to rewrite
Author Response
Reviewer’s comments: Ashraf Madhoun and colleagues submitted a manuscript entitled “Increased Adipose Tissue Expression of Dectin-1 in obesity: Association with Metabolic Inflammation” for publication as an original article in Cells, MDPI. Current study investigated in adipose tissue samples from 59-human subjects (non-diabetic) with obesity. The authors have investigated the expression of Dectin-1 in individuals with obesity and assessed that Dectin-1 transcript expression is positively correlated with higher transcript levels of pro-inflammatory markers related macrophage infiltration, cytokine and chemokines ligands and their receptors. The study design is appropriate, and manuscript written very well, but several issues raise concerns.
Author response: The authors thank the reviewer for understanding our study objectives and for sharing the constructive comments and valuable suggestions for improvement of the manuscript.
Comments:
Reviewer’s comment 1: As manuscript title appears like Adipose tissue by itself express the Dectin-1, as authors also mentioned in the introduction, Dectin-1 is transmembrane protein expressed in monocytes, Macrophages and other inflammatory cells. Therefore, I strongly suggest/recommend to authors to change the title of current manuscript and should be more realistic and meaningful. For Example: Dectin-1 acts as potential Inflammatory Biomarker for metabolic inflammation in Adipose tissue of individuals with Obesity.
Author response 1: We thank the reviewer for kind suggestion, and the suggested title appears to be more appropriate and aligned with the study findings. The title is now revised as suggested, with a slight modification. The new title is: “Dectin-1 as a Potential Inflammatory Biomarker for Metabolic Inflammation in Adipose Tissue of Individuals with Obesity”.
Reviewer’s comment 2: As it known from previous publication (cell reports 2017), MyD88 is negatively corelated with Dectin-1 expression levels. In current study, do you see the regulation of MyD88 is increased or decreased? Authors need to show MyD88 transcript expression in bar graphs together with Figure1.
Author response 2: In our study, we found that adipose dectin-1 expression is positively correlated with the expression of MyD88, which also independently predicted dectin-1 gene expression as per our multi-linear regression analysis. The discrepancies between two studies could be attributable to the pathophysiological differences between humans (our study) and mice (Cell reposts 2017 study). Please see Table 3 and the dot plot distribution (Supplementary Figure 2).
Reviewer’s comment 3: In Figure 1 Legend needs to reformat and rewrite including the Figure 1 title. The description in Figure 1 legend is very confusing because authors start with describing Figure 1A, then suddenly immediately jumped to 1B before even finish describing 1A. Authors need to re-write.
Author response 3: We thank the reviewer for kind suggestion. As advised, we have now modified the legend to Fig 1. We have also included more information in the legend as requested by another reviewer. (Please see page 6-7; lines 228-237).

Round 2
Reviewer 3 Report
The authors addressed most issues raised and answered the questions satisfactorily. The work was improved in its significant part and is acceptable for publication in its present form.